# Image-based phenotyping of cassava roots for diversity studies and carotenoids prediction

**Ravena Rocha Bessa de Carvalho**[1☯], **Diego Fernando Marmolejo Cortes**[2☯],
**Massaine Bandeira e Sousa**[2☯], **Luciana Alves de Oliveira**[2☯], **Eder Jorge de Oliveira**[2☯] *

**1** Centro de Ciências Agrárias, Ambientais e Biológicas, Universidade Federal do Recôncavo da Bahia, Rua Rui Barbosa, Cruz das Almas, BA, Brazil, **2** Embrapa Mandioca e Fruticultura, Rua da Embrapa, Cruz das Almas, BA, Brazil

☯ These authors contributed equally to this work.
* eder.oliveira@embrapa.br

## Abstract

Phenotyping to quantify the total carotenoids content (TCC) is sensitive, time-consuming, tedious, and costly. The development of high-throughput phenotyping tools is essential for screening hundreds of cassava genotypes in a short period of time in the biofortification program. This study aimed to (i) use digital images to extract information on the pulp color of cassava roots and estimate correlations with TCC, and (ii) select predictive models for TCC using colorimetric indices. Red, green and blue images were captured in root samples from 228 biofortified genotypes and the difference in color was analyzed using $L^*$, $a^*$, $b^*$, hue and chroma indices from the International Commission on Illumination (CIELAB) color system and lightness. Colorimetric data were used for principal component analysis (PCA), correlation and for developing prediction models for TCC based on regression and machine learning. A high positive correlation between TCC and the variables $b^*$ (r = 0.90) and chroma (r = 0.89) was identified, while the other correlations were median and negative, and the $L^*$ parameter did not present a significant correlation with TCC. In general, the accuracy of most prediction models (with all variables and only the most important ones) was high ($R^2$ ranging from 0.81 to 0.94). However, the artificial neural network prediction model presented the best predictive ability ($R^2 = 0.94$), associated with the smallest error in the TCC estimates (root-mean-square error of 0.24). The structure of the studied population revealed five groups and high genetic variability based on PCA regarding colorimetric indices and TCC. Our results demonstrated that the use of data obtained from digital image analysis is an economical, fast, and effective alternative for the development of TCC phenotyping tools in cassava roots with high predictive ability.

## Introduction

Cassava (*Manihot esculenta* Crantz), belonging to the Euphorbiaceae family, is a globally cultivated species, due to its advantageous traits such as tolerance to water deficit, with wide

**Data Availability Statement:** All relevant data are within the paper and its Supporting Information files.

**Funding:** • Ravena Rocha Bessa de Carvalho: CAPES (Coordenação de Aperfeiçoamento de Pessoal de Nível Superior). Grant number:88882.424436/2019-01 • Eder Jorge de Oliveira: CNPq (Conselho Nacional de Desenvolvimento Científico e Tecnológico). Grant number:409229/2018-0, 442050/2019-4 and 303912/2018-9 • Eder Jorge de Oliveira: FAPESB (Fundação de Amparo à Pesquisa do Estado da Bahia). Grant number: Pronem 15/2014 • Eder Jorge de Oliveira and Massaine Bandeira e Sousa: UK's Foreign, Commonwealth & Development Office (FCDO) and the Bill & Melinda Gates Foundation. Grant number: INV-007637 • The funder provided support in the form of fellowship and funds for the research, but did not have any additional role in the study design, data collection and analysis, decision to publish, or preparation of the manuscript.

**Competing interests:** The authors have declared that no competing interests exist.

adaptation to different edaphoclimatic conditions and satisfactory yields, even in soils with low fertility [1, 2]. Nigeria, with 59.19 million tons cultivated on 7.21 million hectares, is the world's largest producer, while Brazil ranks fifth with production of 17.49 million tons in an area of 1.19 million hectares, and average yield of 14.70 t ha$^{-1}$ [3].

Cassava roots are basically composed of carbohydrates, making them an important source of energy for millions of people around the world [4]. The composition of the roots includes cyanogenic glycosides (linamarin and lotaustralin), present in concentrations ranging from 9.90 to 3927.0 μg g$^{-1}$ [5–7], and low nutritional quality, with considerably reduced values of proteins, minerals (Zn, Fe, N, Ca, P, K, and Mg) and vitamins (B1, B2, B3, C, and β-carotene) [7, 8].

The increase in nutritional content through biofortification is a strategy that has been adding nutritional value to cassava varieties [9]. Biofortification is an international and interdisciplinary initiative that seeks to reduce human malnutrition by increasing the concentration of micronutrients in various staple crops, such as increasing the carotenoids content in cassava [9, 10]. In cassava roots, the aim is to increase the β-carotene content (precursor of vitamin A), with the main purpose of combating hidden hunger caused by vitamin A deficiency [9, 11]. In addition, biofortification programs seek to adjust the levels of other attributes such as high agronomic performance [12], low levels of cyanogenic compounds and high levels of dry matter in the roots, in order to meet the preferences for sweet cassava market [5, 9].

Despite important advances in increasing the carotenoids content obtained in several conventional breeding programs [5, 13, 14], phenotyping to quantify the total carotenoids content (TCC) or β-carotene is quite sensitive, time-consuming, tedious, and costly. Therefore, new rapid phenotyping methods are needed, since biofortification programs require the screening of hundreds of samples in a short period of time [5, 10].

Recently, predictions based on near-infrared spectroscopy (NIRS) have demonstrated high potential for indirect phenotyping of carotenoid content in cassava [5, 11, 15–17]. In addition, methodologies based on digital image analysis allow extracting color information [18] due to the strong correlation between digital and visual data [19]. Since the present carotenoids have a strong correlation with the intensity of the yellow color [14], it is assumed that this type of phenotyping is a viable and adequate option for the quantification of the carotenoid content.

The main advantages of image-based root phenotyping are savings time, use of commercially available digital cameras that are easy to handle, transport and use open-source software for image processing, demonstrating high potential for use in plant breeding programs [19. 20]. The recorded images also allow researchers to go back and re-examine them whenever doubts arise about the phenotyping process [18]. Furthermore, once the prediction models are calibrated, it is possible to reduce the number of laboratory samples, concentrating efforts only on those of greatest interest, and thus, greatly reducing the cost of the analyses necessary for the selection of genotypes.

Rapid imaging phenotyping has already been widely used to study the relationship between color indices extracted from images and carotenoid content in different crops such as carrots (*Daucus carota*) [21], tomatoes (*Solanum lycopersicum* L.) [22], olive oil (*Olea europaea*) [23] and mandarin (*Citrus reticulata* Blanco) [24]. Typically, images are acquired by a digital device and saved in the three-dimensional red, green and blue (RGB) color space; however, this space is not uniform in percentage and does not represent colors naturally perceived by human vision [25]. Some studies have opted to convert RGB color matrices into International Commission on Illumination (CIELAB) codes [26], to correlate image colors with some specific attribute in plants [27–29]. The CIE [26] color model is a predominant choice among researchers [25] because it has a color space with a standardized measurement technique [30]. The CIELAB color space [26] proved to be quite accurate in estimating the carotenoids content

based on digital images in tomato (*Solanum lycopersicum*) and in pollen of species from the Brassicaceae, Myrtaceae and Fabaceae families [27, 31].

In other species, high predictive ability for traits associated with nutritional quality has been observed when using colorimetric indices obtained from digital images. For example, in mandarin (*Citrus reticulata* Blanco), partial least squares ($R^2$ = 0.96) and third-order linear regression ($R^2$ = 0.95) models were able to predict fruit maturation (citrus color index) with high precision [24]. In olive oil (*Olea europaea*), the support vector machine ($R^2$ = 0.96) and artificial neural networks ($R^2$ = 0.94) models also showed high predictive ability for carotenoid content in this species [23]. Additionally, a multiple linear regression model was able to predict the carotenoids content with high precision ($R^2$ = 0.89) in pollen of species from the Brassicaceae, Myrtaceae, Fabaceae families [31].

Considering the need to develop high-performance phenotyping methodologies to optimize the selection of biofortified sweet cassava in the breeding programs, the present study aimed to: (i) evaluate the potential of using digital images to capture information on the pulp color of cassava roots and estimate correlations with TCC; (ii) select predictive models powered by colorimetric indices that have high precision and predictive ability to obtain TCC.

## Material and methods

### Plant material and experimental design

The calibration set was composed of 228 genotypes from the Cassava Germplasm Bank of Embrapa Mandioca e Fruticultura, Cruz das Almas, Bahia, Brazil (Latitude: 12˚ 40' 12" S, Longitude: 39˚ 06' 07" W and Altitude: 220m). This germplasm panel is composed of genotypes with high diversity for carotenoid content, cyanogenic compounds, dry matter content of the root and several yield attributes. The clones were cultivated from June 2019 to July 2020 at the Experimental Station of Embrapa Cassava e Fruticultura, using an augmented block design, with 34 local and improved varieties as common controls. The experimental plot consisted of two lines with 10 plants each (20 plants per plot), with a spacing of 0.90 m between lines and 0.80 m between plants. Planting was carried out with 16 cm stakes in rainfed condition, following the recommendations and agricultural practices for cassava crop [32]. To compose the calibration set, clones with cream and yellow root pulp coloration were previously selected.

The climate in the germplasm growing region is tropical, hot, and humid, without a dry season. With an average annual rainfall of 1,170 mm, the wettest months are from March to August and the driest ones from September to February. The average annual temperature is 24.5˚C, with an average of 80% relative humidity. The predominant soils in the region are Dystrophic Yellow Latosols.

### Obtaining and processing images

A box (dimensions 24×18×19 cm), with white inner walls and the bottom lined with blue plastic material to increase the precision of the separation of the pixels from the root, was used as a phenotyping platform. We used 6500k frequency (Cool White Light) 9W lamps, placed equidistantly inside the box. To obtain the digital images, a tablet with an Android 9.0 system and an 8MP camera was attached to the top of the box. The digital images were captured in an environment with controlled temperature (22˚C±2˚C) to prevent carotenoid degradation.

During harvest, 4–6 commercial roots (>5 cm in diameter and around 12–20 cm in length) of all 228 genotypes were selected. The roots were washed with running water to remove excess adhered soil; the central region was then cut into pieces approximately 2 cm thick. Six pieces of root per genotype were placed in the center of the camera's field of view and three RGB images were captured. Images were saved in Joint Photographic Experts Group (JPEG) format

with a resolution of 2448 × 3264 pixels. The images were pre-processed using Gaussian Blur filters to remove the background with the aid of ImageJ software version 1.52a. Colorimetric analysis was performed using the Tomato Analyzer–Color Test version 4.0 software [30]. The initial values of the chromatic components ($L^*$, $a^*$ and $b^*$) of the CIELAB system [26] were obtained using the standard calibration of the software. This calibration was performed assuming values of 1.0 for the slope and 0 for the intercepts of a linear regression equation. The excess root matter was processed with the aid of a food multiprocessor and reserved for TCC analysis.

## Root color evaluation

The differences in root color between the different cassava genotypes were evaluated using the CIELAB parameters [26], which were obtained from the RGB values. The RGB conversion into L*, a* and b* was performed in three steps [30].

1. Initially, the RGB values were scaled to a uniform color space using the following equations:

$$Var\_R = (\left\{ \left[ (\frac{R}{255}) + 0.055 \right] \right\}^{2.4})/1.055) \times 100$$

$$Var\_G = (\left\{ \left[ (\frac{G}{255}) + 0.055 \right] \right\}^{2.4})/1.055) \times 100$$

$$Var\_B = (\left\{ \left[ (\frac{B}{255}) + 0.055 \right] \right\}^{2.4})/1.055) \times 100.$$

2. The scaled RGB values were converted to XYZ tristimulus values using the following equations:

$$X = (Var\_R \times 0.4124) + (Var\_G \times 0.3576) + (Var\_B \times 0.1875)$$

$$Y = (Var\_R \times 0.2126) + (Var\_G \times 0.7152) + (Var\_B \times 0.0722)$$

$$Z = (Var\_R \times 0.0193) + (Var\_G \times 0.1192) + (Var\_B \times 0.9505)$$

3. The XYZ values were converted to L^*, a^*, and b^* values using the following equations:

$$L^* = 116f\left(\frac{Y}{Yn}\right) - 16$$

$$a^* = 500[f\left(\frac{X}{Xn}\right) - f(\frac{Y}{Yn})]$$

$$b^* = 200[f\left(\frac{Y}{Yn}\right) - f(\frac{Z}{Zn})]$$

$$\text{where} f(q) = \begin{cases} q^{1/3} & \text{if } q > 0.008856 \\ 7.787q + \dfrac{16}{116} & \text{if } q \leq 0.008856 \end{cases};$$

*Yn*, *Xn*, and *Zn* represent the tristimulus values from illuminant and observer angles; and for D65, observer angle 10, *Xn* = 94.83, *Yn* = 100.0, and *Zn* = 107.38.

The $L^*$ coordinate indicates the lightness, from dark (0) to light (100), of the colour, while $a^*$ and $b^*$ indicate the variation from green to red (-60 to 60) and from blue to yellow (-60 to 60), respectively. For each sample, each record represents an average of six measurements. The cylindrical coordinates, chroma and hue, were calculated from $a^*$ and $b^*$. Hue is an angular measurement between 0˚ and 360˚ that represents the base color. Its value was calculated based on the following equations:

$$Hue = \frac{180}{pi} \times cos\left[a/\sqrt{(a*)^2 + (b*)^2}\right] \text{ for } a* > 0$$

$$360 - \left\{\frac{180}{pi} \times cos\left[a/\sqrt{(a*)^2 + (b*)^2}\right]\right\} \text{ for } a* < 0.$$

The chroma value is the relative saturation, represented by the mean of $a^*$ and $b^*$. The chroma was calculated using the following equation:

$$chroma = \sqrt{(a*)^2 + (b*)^2}$$

The lightness was also calculated, defined by the average amount of lightness (ranging from 0 to 240) in all pixels. Luminance accounts for the human eye's varying sensitivity to radiation at various wavelengths and defines brightness. The lightness was estimated from the RGB value of each pixel using the expression:

$$Lightness = \frac{[maximum(R, G, B) + minimum(R, G, B) \times 240]}{2 \times 255}$$

### Total carotenoid content analysis

After capturing the root imagens, two samples were collected for the TCC analysis, each containing 10 g, 15 g or 25 g of the crushed roots (depending on the intensity of the pulp color), in addition to a backup sample of 60 g. The samples were placed in glass jars with lids and covered with aluminum foil to avoid contact with light as much as possible, to prevent degradation of the carotenoids. The samples were then frozen for further analysis.

To quantify the carotenoid content, the procedure described by the HarvestPlus Handbook for Carotenoid Analysis [33] was adopted. The pigments from the crushed roots were extracted by grinding the samples in an Ultra Turrax homogenizer for three minutes, in order to break down the plant cells to then allow the carotenoids' extraction with approximately 50 mL of acetone. The sample and acetone mixture were filtered through a Buchner funnel with the aid of a vacuum pump, and the residue retained in the funnel was washed with acetone until it did not show any color. The extract containing only the pigment and acetone was reserved in the suction flask, then transferred to a separating funnel containing petroleum ether (of which the quantity varied depending on the color of the sample); approximately 250mL of saline solution was added slowly to induce phase separation. The aqueous phase was

discarded and the saline washing procedure repeated five times until only petroleum ether and pigment remained. This new extract was filtered through a funnel with glass wool and anhydrous sodium sulfate into an amber volumetric flask, where its volume was supplemented by petroleum ether. An aliquot of each sample was taken for determination by a spectrophotometer (UV-Vis Thermo Scientific, Genesis 10S model), adjusted for an absorbance at 450 nm.

The TCC was calculated using the following formula:

$$\text{TCC}(ug.g^{-1}) = \frac{A \times V(ml) \times 10^4}{A_{1cm}^{1\%} \times P(g)},$$

where $A$ is the absorbance, $V(ml)$ is the total extract volume in milliliters, $P(g)$ is the sample weight in grams, and $A_{1cm}^{1\%}$ is equal to the extinction coefficient of β-carotene in petroleum ether (2592).

## Data analysis

Initially, Pearson's correlation analysis was performed between the phenotypic data from the TCC and the colorimetric data from the CIE color system in order to verify the magnitude and direction of the correlation between these parameters. PCA was implemented to verify the phenotypic diversity of genotypes based on TCC and colorimetric indices, as well as to verify the importance of colorimetric indices as predictors of TCC. The grouping of genotypes was determined by the K-means clustering algorithm. These analyses were all performed using the factoextra package [34] implemented in the R version 4.03 programming environment [35].

Then, the colorimetric data were used to develop models to predict TCC. Twelve predictive models derived from regression and machine learning were tested: Linear Regression with Forward Selection (LRFS), Linear Regression with Backwards Selection (LRBS), Ridge Regression (RR), Linear Regression with Stepwise Selection (LRSS), Generalized Linear Model with Stepwise Feature Selection (GLMSS), Random Forest (RF), Partial Least Squares (PLS), the Bayesian Lasso (BL), the Bayesian Blasso (BBL), Artificial Neural Network (ANN), Support vector machine (SVM), and Classification and regression trees (CART) [36, 37]. The predictive models were implemented in the caret package (Kuhn, 2008) from R software version 4.03 [35].

To test the accuracy of the predictive models, we used different cross-validation schemes: 1) random cross-validation without test set (V-Random), where genotypes were randomly split 80/20% into training and validation, respectively; 2) clustering-based, where PCA clustering with k = 5 (IV-Cluster) clusters were assigned to the split sets in order to have a group of clusters in training, validation and testing with no further criteria; respectively; and 3) random cross-validation with test set (IV-Random): where genotypes were randomly split 60/20/20% into training, validation, and test sets respectively. For the EV-Cluster strategy, we considered the population structure of genotypes determined by PCA and k-means clustering. The training and validation sets were composed of genotypes from all clusters, except the cluster used as a test set. For example, in the independent validation of the first cluster, the accessions belonging to the second, third, fourth and fifth clusters were used for training and validation. In all cross-validation schemes, the training and validation sets were formed considering 5-fold increase and 6 repetitions. The process was randomly repeated 30 times.

The performance of the predictive models was evaluated based on the root-mean-square error (RMSE) and the coefficient of determination ($R^2$) values, obtained in each cross-validation partition. The RMSE measures the average magnitude of the estimated errors; the closer its positive value is to 0, the higher the quality of the estimated values. It is calculated according

to the following equation:

$$RMSE = \sqrt{\frac{1}{n}\sum_{i=l}^{n}(E_i - O_i)^2},$$

where $E_i$ and $O_i$ are the estimated and observed values, respectively, and $n$ is the number of observations. By contrast, $R^2$ varies between 0 and 1, and the closer to 1, the better its explanatory power. $R^2$ is described as the ratio that represents the proportion of the total variation of the dependent variable that is explained by the variation of the independent variable. It is estimated by the following equation:

$$R^2 = \frac{\sum_{i=l}^{n}(\widehat{y}_i - \overline{y})^2}{\sum_{i=l}^{n}(y_i - \overline{y})^2},$$

where $\sum_{i=l}^{n}(\widehat{y}_i - \overline{y})^2$ corresponds to the explained variation, and $\sum_{i=l}^{n}(y_i - \overline{y})^2$ represents the variation not explained by the model.

The mean absolute percentage error (MAPE) was also used to estimate the accuracy of the TCC predictions. MAPE was estimated using the following equation:

$$MAPE = \frac{1}{n}\sum_{t-1}^{n}|\frac{y_t - \widehat{y}_t}{y_t}|$$

where $y_t$ is the observed value, $\hat{y}_t$ is the predicted value, and $n$ is the number of fitted values.

## Results

### Correlation between total carotenoid content and colorimetric indices

The TCC ranged from 0.30 μg g$^{-1}$ to 13.67 μg g$^{-1}$ based on the fresh weight of the samples, while the overall mean of the 228 cassava genotypes was 3.56 μg g$^{-1}$. Scatter plots were constructed to assess the distribution of data and estimate the Pearson correlation coefficients between the TCC and the colorimetric indices extracted from the digital images (Fig 1, S1 Table). A strong positive correlation was identified between TCC and the variables $b^*$ (r = 0.90) and chroma (r = 0.89), while moderate and negative correlations were identified between TCC and the indices $a^*$ (r = -0.53), hue (r = -0.59), and lightness (r = -0.66). On the other hand, the $L^*$ parameter was not found to be significantly correlated with TCC.

### Phenotypic diversity and clustering of cassava genotypes based on total carotenoid content and colorimetric indices

Population structuring based on the TCC dataset and colorimetric indices was evaluated using PCA. According to the K-means clustering algorithm, five groups were identified based on the sum of squares within groups (Fig 2).

The first two main components explained 89% of the total variance of the sample population dataset and, therefore, allow a reasonable representation of the cassava genotype clustering based on the evaluated parameters (Fig 3). There was wide phenotypic variation among cassava genotypes for both TCC and colorimetric indices. The TCC and the colorimetric indices $b^*$ and chroma were the variables that contributed most to the dispersion of the genotypes. Similarly to the findings of the correlation analyses, the $b^*$ and chroma indices were those that showed the greatest association with TCC, and therefore have high relative importance for the prediction of this phenotype. On the other hand, the $L^*$ index contributed the least to the phenotypic variation of the cassava genotypes.

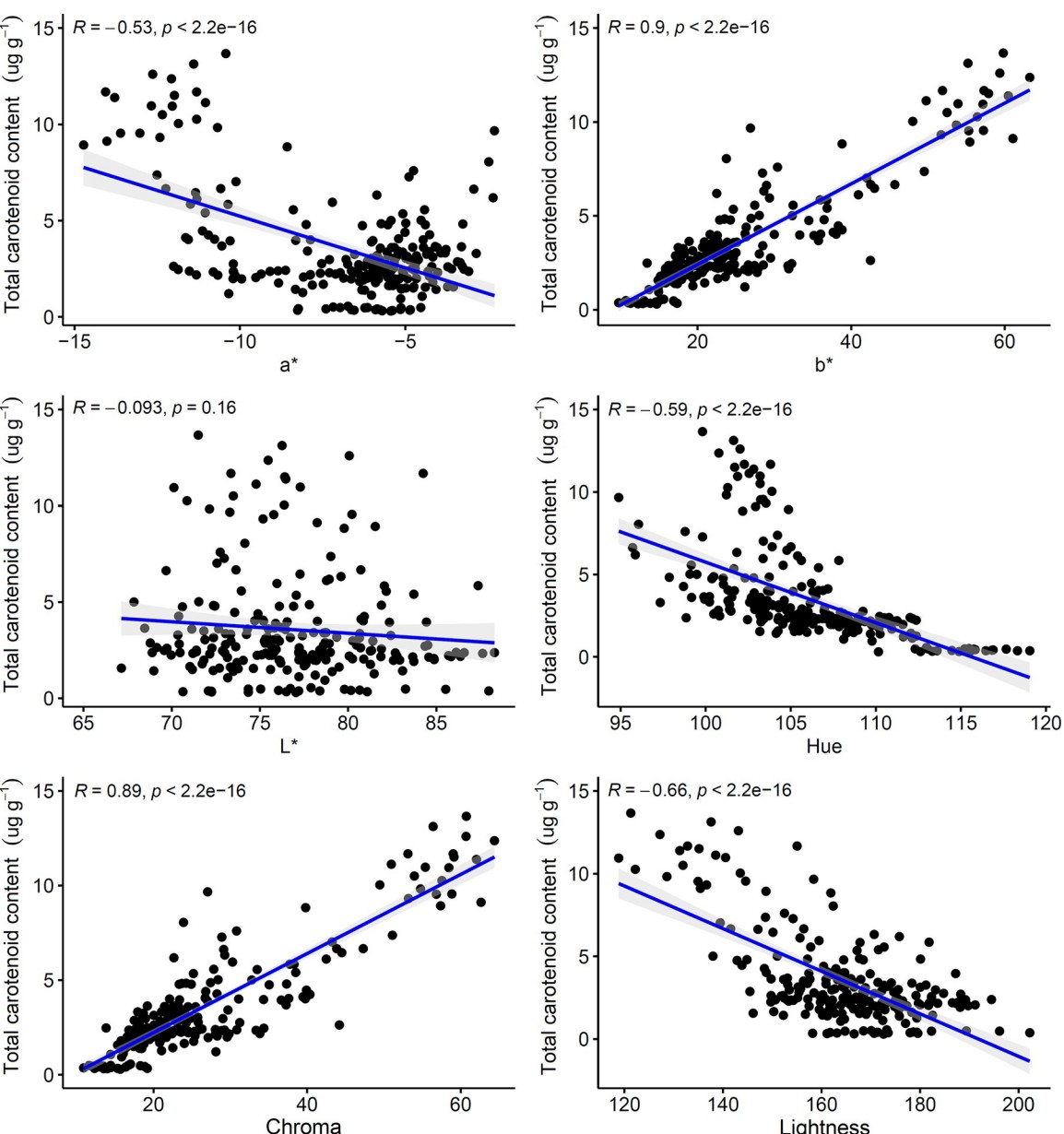

**Fig 1. Scatter plots of the total carotenoid content and colorimetric indices.** Numerical values represent Pearson's correlation coefficients between colorimetric indices and total carotenoid content. The blue line represents the 1:1 isoline.

Based on PCA, the cassava genotypes were categorized into five groups. Groups 1, 2, 3, 4 and 5 comprised 76, 34, 26, 21 and 71 genotypes, respectively. Groups 1 and 2 consisted of genotypes with low TCC values (1.91 µg g⁻¹ and 1.82 µg g⁻¹, respectively) and low $b^*$ and chroma, in addition to the highest $L^*$, lightness and hue (Fig 4). Groups 3 and 5 consisted of cassava genotypes with yellow pulp, but with moderate TCC contents (4.82 µg.g⁻¹ and 3.60 µg. g⁻¹, respectively) and moderate lightness values (varying from 158 to 159). However, these two groups differed in their values of $a^*$ (-10.23 and -4.60 in groups 3 and 5, respectively), $b^*$ (36.48 and 22.20), $L^*$ (78.60 and 73.10), hue (105.59 and 101.87), and chroma (37.93 and 22.72).

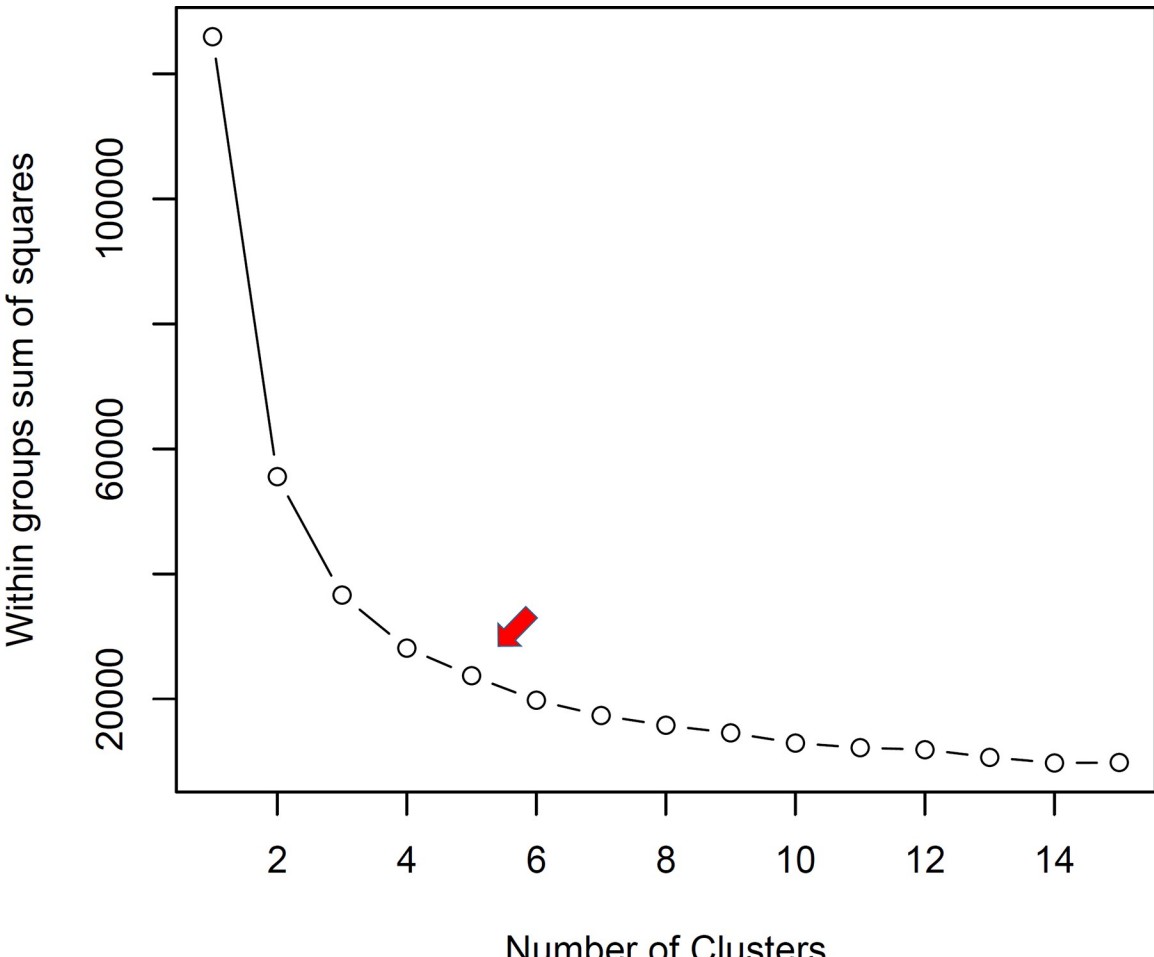

**Fig 2. Within-group sum of squares of 228 biofortified cassava genotypes based on phenotypic data of total carotenoid content and colorimetric indices (CIELAB).** The grouping criterion was based on the K-means clustering algorithm.

Group 4 was composed of cassava genotypes with the highest TCC (mean 10.74 μg g$^{-1}$). Regarding the means of the colorimetric indices, group 4 was characterized by high chroma (56.96) and $b^*$ (55.57); low hue values (102.59), $a^*$ (-12.38), and lightness (136.04); and moderate values of $L^*$ (76.14).

## Prediction of total carotenoid content using colorimetric indices

The performance of predictive models for TCC in cassava roots using colorimetric data obtained from digital images was evaluated considering the complete model (all variables used for TCC prediction) and the reduced model, in which only the variables with relative importance greater than or equal to 50% for the TCC prediction were retained in the different models evaluated.

The colorimetric indices $b^*$ and chroma, each with a relative importance of 100%, were the variables that contributed most to the prediction of TCC in the majority of models (Fig 5). The exception was the PLS model, in which the $b^*$ and $a^*$ indices each had a relative importance of 100%. The lightness and hue indices achieved importance between 50% and 70% depending on the model under consideration. For all models, the $L^*$ index was below the 50% importance threshold and was hence excluded from the reduced models. Therefore, most of the reduced

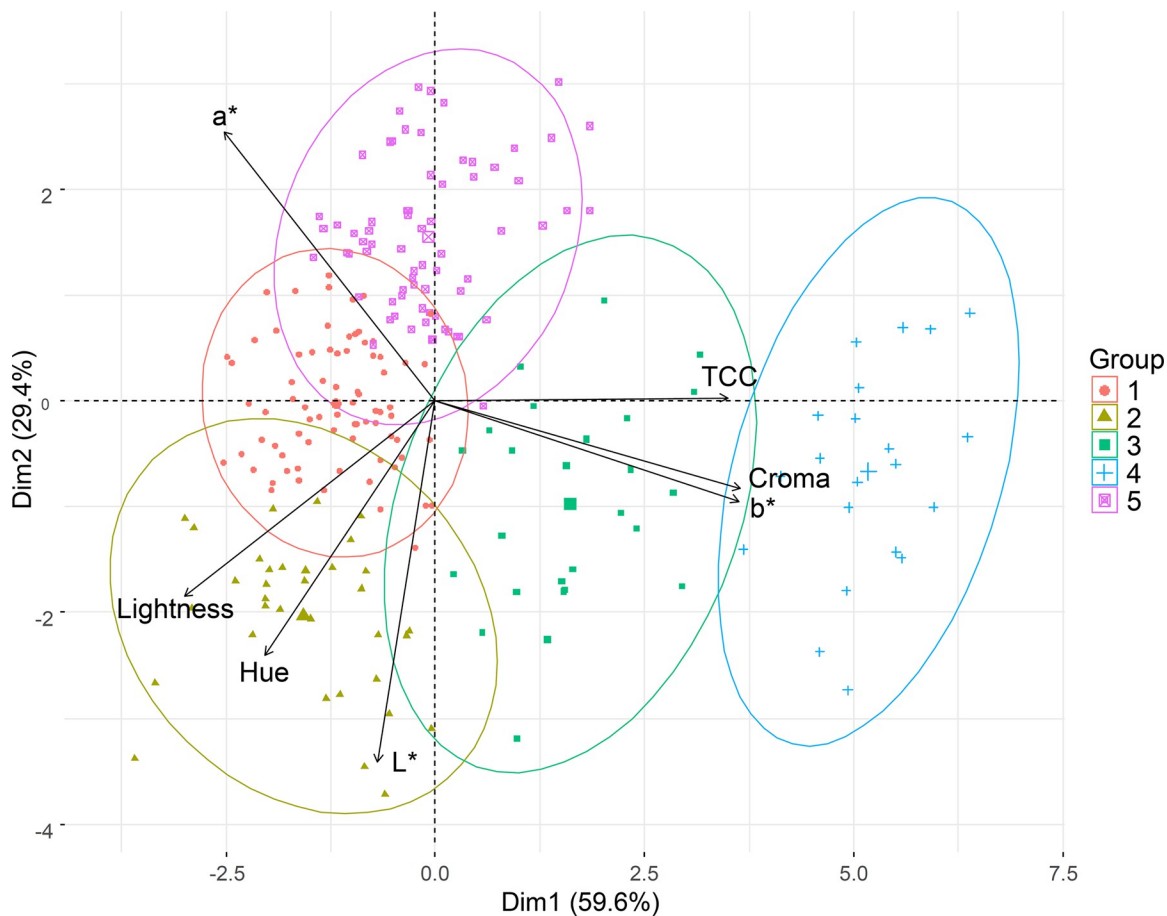

**Fig 3. Principal Component Analysis (PCA) based on phenotypic data of total carotenoid content and colorimetric indices (CIELAB) evaluated in 228 biofortified cassava genotypes.**

models included the variables $a^*$, $b^*$, chroma, hue, and lightness, with the exception of the RF, whose reduced model was represented only by $b^*$, hue and chroma. Despite the reduced models containing only the variables with the greatest predictive ability for TCC, there were no differences in their predictive abilities compared to the complete model (Table 1; S1 Fig). Only the RF and PLS models showed a small difference in prediction accuracy, with $R^2 = 0.91$ and 0.90 (RF) and $R^2 = 0.94$ and 0.93 (PLS) in the complete and reduced models, respectively.

Regardless of the cross-validation strategy, the predictive ability between the complete and reduced models was similar ($p > 0.05$) for all prediction models, showing that the selection of more important features did not negatively affect the model's performance (Table 1; S1 Fig). The error associated with TCC prediction was low ($0.04 \leq MAPE \leq 0.13$) in most models (Table 1) using V-Random cross-validation. The BL, BBL and LRFS models had the lowest MAPE values (0.04) using the reduced model, while the CART model had higher MAPE (0.13) in both the complete and reduced models. In general, the use of reduced models did not negatively affect the prediction accuracy of the TCC values considering that MAPE values were less than or equal to those obtained for most complete models. In the BBL, BL, LRFS and SVM models, there was a slight improvement in accuracy when using the reduced model compared to the complete model.

In general, the accuracy of the predictive models (both complete and reduced) was high, with $R^2$ ranging from 0.90 (SVM) to 0.94 (ANN) (Table 1, S1 Fig). Only the CART predictive

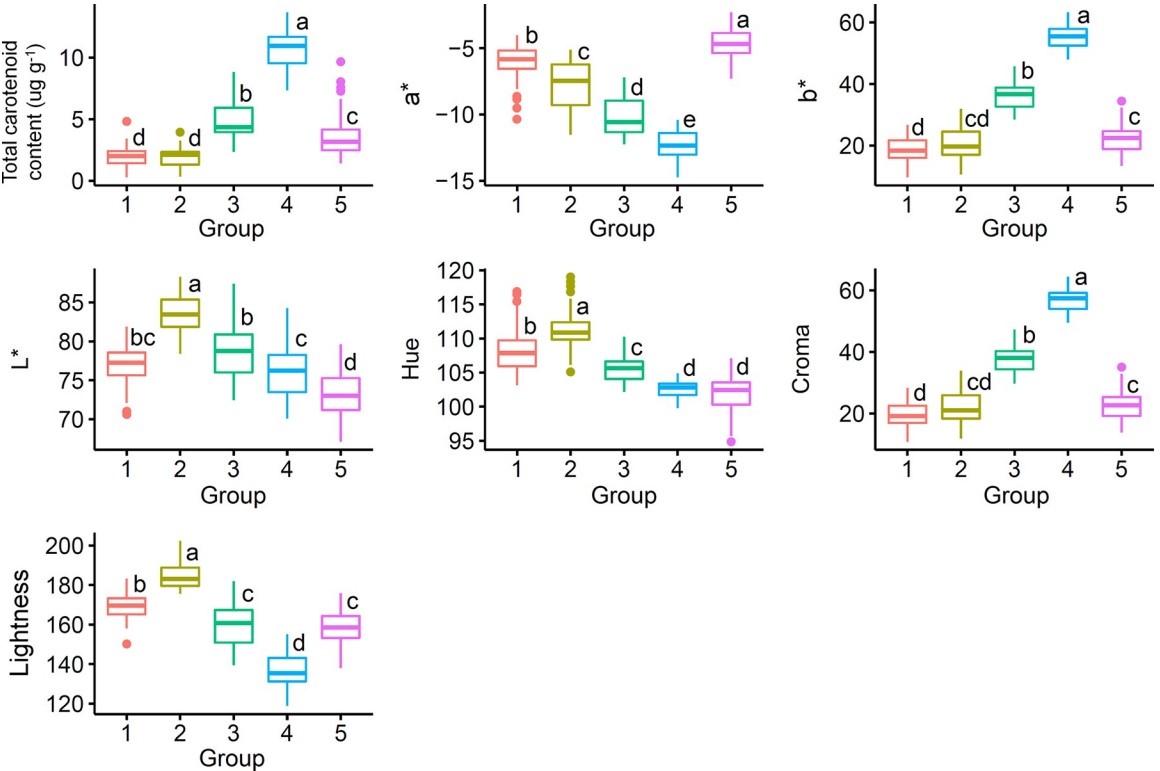

**Fig 4. Boxplot of total carotenoid content and colorimetric indices for each cluster of 228 biofortified cassava genotypes based on principal component analysis.** Different letters represent significant differences between accession groups with $p < 0.05$ by the Tukey Honest Significant Difference test.

model produced a TCC prediction accuracy below 0.90. In contrast, the ANN model had the best predictive ability ($R^2 = 0.94$), associated with the lowest error in the TCC estimates (RMSE = 0.24). The models based on linear regression (LRFS, LRBS, LRSS, and GLMSS), as well as the PLS, BL, BBL, and RR models, performed similarly (RMSE = 0.26; $R^2 = 0.93$). Although the RF and SVM models had high predictive abilities ($R^2 = 0.90$ for both complete and reduced models), they also had higher RMSE values (0.31 and 0.32 for complete and reduced model, respectively (RF) and 0.33 for both complete and reduced (SVM).

The performance of two additional cross-validation schemes with an independent sample was also analyzed to provide an unbiased evaluation of the goodness fit of the training set. The IV-Cluster and IV-Random routines were performed using the complete and reduced models (Table 1, S2 and S3 Figs). In general, IV-Random exhibited higher accuracy than IV-Cluster. The accuracy of IV-Random considering the complete and reduced models was similar (0.90 and 0.94), except for the model CART, which had the lowest $R^2$ value (0.81). On the other hand, the $R^2$ value to the cross-validation scheme (IV-Cluster) ranged from 0.23 (SVM) to 0.71 (LRFS, LRBS, RR, PLS and BBL).

For the IV-Cluster, the SVM model had the lowest accuracy for both complete ($R^2 = 0.23$) and reduced models ($R^2 = 0.29$). Regarding the RMSE, the prediction models of the V-Random and IV-Random strategies showed a best fit for both complete and reduced models (range from 0.24 to 0.43), while the IV-Cluster was the worst one (range from 0.29 to 0.97) (Table 1; S2 and S3 Figs). The cross-validation scheme IV-Cluster had lower MAPE values compared to IV-Random and V-Random (Table 1). The MAPE values of the IV-Random were quite similar between the complete and reduced models ($0.05 \leq MAPE \leq 0.14$), while

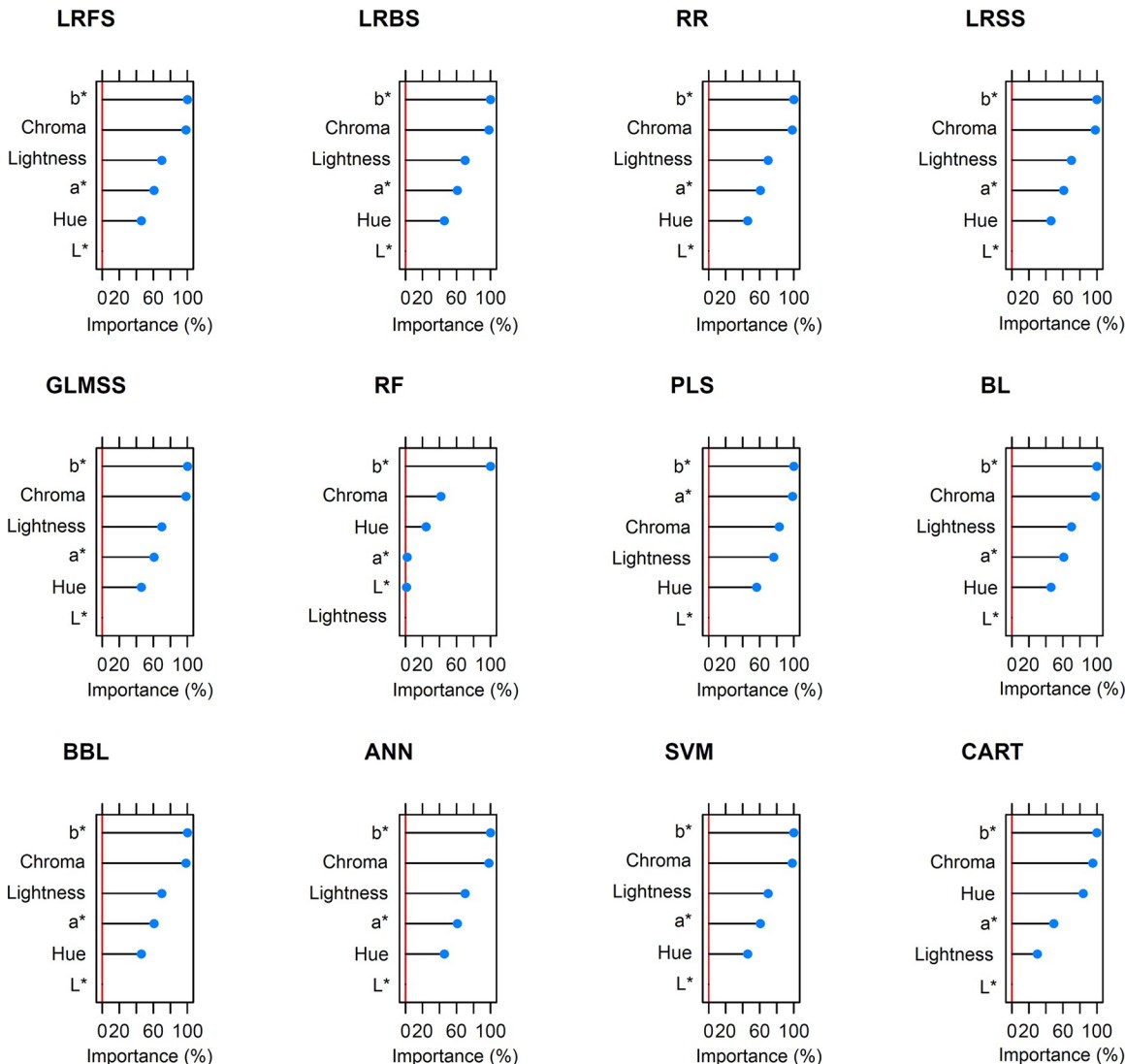

**Fig 5. Relative importance of colorimetric indices for predicting total carotenoid content using 12 prediction models: Linear Regression with Forward Selection (LRFS), Linear Regression with Backwards Selection (LRBS), Ridge Regression (RR), Linear Regression with Stepwise Selection (LRSS), Generalized Linear Model with Stepwise Feature Selection (GLMSS), Random Forest (RF), Partial Least Squares (PLS), the Bayesian Lasso (BL), the Bayesian Blasso (BBL), Artificial Neural Network (ANN), Support vector machine (SVM), and Classification and regression trees (CART).**

for the IV-Cluster strategy there was only a small MAPE reduction in the reduced model ($0.14 \leq$ MAPE $\leq 0.77$) compared to the complete model ($0.13 \leq$ MAPE $\leq 0.92$) for some prediction models such as RF and SVM.

For both V-Random and IV-Random schemes, the scatterplot of observed and predicted TCC values clearly shows the good fit of the ANN model ($R^2 = 0.94$) followed by the BBL, BL, GLMSS, LRBS, LRFS, LRSS, PLS, and RR models ($R^2 = 0.93$) regardless of the fold validation (Figs 6 and 7). On the other hand, the CART ($R^2 = 0.81$ to $0.82$), RF and SVM ($R^2 = 0.90$) models exhibited greater dispersion of the predicted data and, therefore, less adjustment for the TCC prediction in cassava roots. In general, for most models, the TCC predictions with values between 0.50 and 5.0 µg g$^{-1}$ were reasonably accurate. However, above this value, the

**Table 1. Performance of different prediction models for total carotenoid content in cassava roots using colorimetric indices obtained from digital images considering the complete model (all variables) and reduced model (variables with more than 50% relative importance), using the random cross-validation without test set (V-Random), PCA clustering-based cross-validation (IV-Cluster), and random cross-validation with test set (IV-Random).**

| Model[a] | V-Random | | | | | | IV-Cluster | | | | | | IV-Random | | | | | |
|---|---|---|---|---|---|---|---|---|---|---|---|---|---|---|---|---|---|---|
| | Complete model | | | Reduced model | | | Complete model | | | Reduced model | | | Complete model | | | Reduced model | | |
| | $R^{2b}$ | $RMSE^c$ | $MAPE^d$ | $R^2$ | RMSE | MAPE | $R^2$ | RMSE | MAPE | $R^2$ | RMSE | MAPE | $R^2$ | RMSE | MAPE | $R^2$ | RMSE | MAPE |
| LRFS | 0.93 | 0.26 | 0.05 | 0.93[ns] | 0.26 | 0.04 | 0.71 | 0.33 | 0.14 | 0.71[ns] | 0.31 | 0.14 | 0.93 | 0.26 | 0.05 | 0.93[ns] | 0.26 | 0.05 |
| LRBS | 0.93 | 0.26 | 0.05 | 0.93[ns] | 0.26 | 0.05 | 0.71 | 0.34 | 0.16 | 0.71[ns] | 0.31 | 0.15 | 0.93 | 0.26 | 0.06 | 0.93[ns] | 0.26 | 0.05 |
| RR | 0.93 | 0.26 | 0.05 | 0.93[ns] | 0.26 | 0.05 | 0.71 | 0.29 | 0.13 | 0.71[ns] | 0.31 | 0.15 | 0.93 | 0.26 | 0.05 | 0.93[ns] | 0.26 | 0.05 |
| LRSS | 0.93 | 0.26 | 0.05 | 0.93[ns] | 0.26 | 0.05 | 0.69 | 0.29 | 0.13 | 0.71[ns] | 0.31 | 0.15 | 0.93 | 0.26 | 0.05 | 0.93[ns] | 0.26 | 0.05 |
| GLMSS | 0.93 | 0.26 | 0.05 | 0.93[ns] | 0.26 | 0.05 | 0.69 | 0.29 | 0.13 | 0.71[ns]. | 0.31 | 0.15 | 0.93 | 0.26 | 0.05 | 0.93[ns] | 0.26 | 0.05 |
| RF | 0.90 | 0.31 | 0.07 | 0.90[ns] | 0.32 | 0.07 | 0.52 | 0.58 | 0.23 | 0.54[ns] | 0.57 | 0.20 | 0.90 | 0.31 | 0.08 | 0.90[ns] | 0.31 | 0.08 |
| PLS | 0.93 | 0.26 | 0.05 | 0.93[ns] | 0.26 | 0.05 | 0.71 | 0.30 | 0.13 | 0.71[ns] | 0.31 | 0.15 | 0.93 | 0.25 | 0.05 | 0.93[ns] | 0.25 | 0.05 |
| BL | 0.93 | 0.26 | 0.05 | 0.93[ns] | 0.26 | 0.04 | 0.66 | 0.30 | 0.14 | 0.66[ns] | 0.32 | 0.17 | 0.93 | 0.26 | 0.05 | 0.93[ns] | 0.26 | 0.05 |
| BBL | 0.93 | 0.26 | 0.06 | 0.93[ns] | 0.26 | 0.04 | 0.71 | 0.32 | 0.15 | 0.71[ns] | 0.32 | 0.15 | 0.93 | 0.26 | 0.05 | 0.93[ns] | 0.26 | 0.05 |
| ANN | 0.94 | 0.24 | 0.10 | 0.94[ns] | 0.24 | 0.10 | 0.66 | 0.39 | 0.33 | 0.68[ns] | 0.46 | 0.31 | 0.94 | 0.24 | 0.07 | 0.94[ns] | 0.24 | 0.07 |
| SVM | 0.90 | 0.33 | 0.10 | 0.90[ns] | 0.33 | 0.09 | 0.23 | 0.97 | 0.92 | 0.29[ns] | 0.91 | 0.77 | 0.90 | 0.32 | 0.11 | 0.90[ns] | 0.32 | 0.09 |
| CART | 0.82 | 0.43 | 0.13 | 0.82[ns] | 0.43 | 0.13 | 0.29 | 0.80 | 0.22 | 0.29[ns] | 0.80 | 0.22 | 0.81 | 0.43 | 0.14 | 0.81[ns] | 0.43 | 0.14 |

[a]: Linear regression with forward selection (LRFS), linear regression with backward selection (LRBS), ridge regression (RR), linear regression with stepwise selection (LRSS), generalized linear model with stepwise feature selection (GLMSS), random forest (RF), partial least squares (PLS), Bayesian Lasso (BL), Bayesian Blasso (BBL), artificial neural network (ANN), support vector machine (SVM), and classification and regression trees (CART).

[b]: $R^2$: coefficient of determination.

[c]: RMSE: root-mean-square error.

[d]: MAPE: mean absolute percentage error.

[ns]: not significant by paired $t$-test.

regression graphs show a greater dispersion of data. One possible explanation is the observation that 82% of genotypes had a TCC below 5.0 µg g$^{-1}$, thus representing most of the samples.

In general, the IV-Cluster scheme exhibited the worst prediction parameters regardless the model used (Table 1, Figs 8 and 9). The best prediction fits were identified in the BBL, LRBS, LRFS, PLS, and RR models (average $R^2$ = 0.71), followed by the GLMSS and LRSS models (average $R^2$ = 0.69). On the other hand, the SVM and CART models presented predictions below 0.30 with the greatest dispersion in the predicted values and the worst prediction adjustment.

In the IV-Cluster scheme, there was wide accuracy variation of the models for predicting the 5 PCA-clusters (Figs 8 and 9). The worst prediction scenario occurred when the genotypes of clusters 1, 2, 3 and 5 were used as the training population (low to medium TCC contents) for the prediction of cluster 4 genotypes (genotypes with high TCC), regardless of the prediction model used. This was possibly due to the fact that the information for predicting genotypes with more than 5 ug g$^{-1}$ was not incorporated in the training population, thus limiting the predictive capacity in the test population.

## Discussion

### Variation and correlation between total carotenoid content and colorimetric indices

The TCC in the 228 cassava genotypes analyzed ranged from 0.30 µg g$^{-1}$ to 13.67 µg g$^{-1}$. This variation in the TCC is consistent with other reports on cassava, such as the screening of 1789 cassava genotypes in the International Center for Tropical Agriculture (Colombia), whose

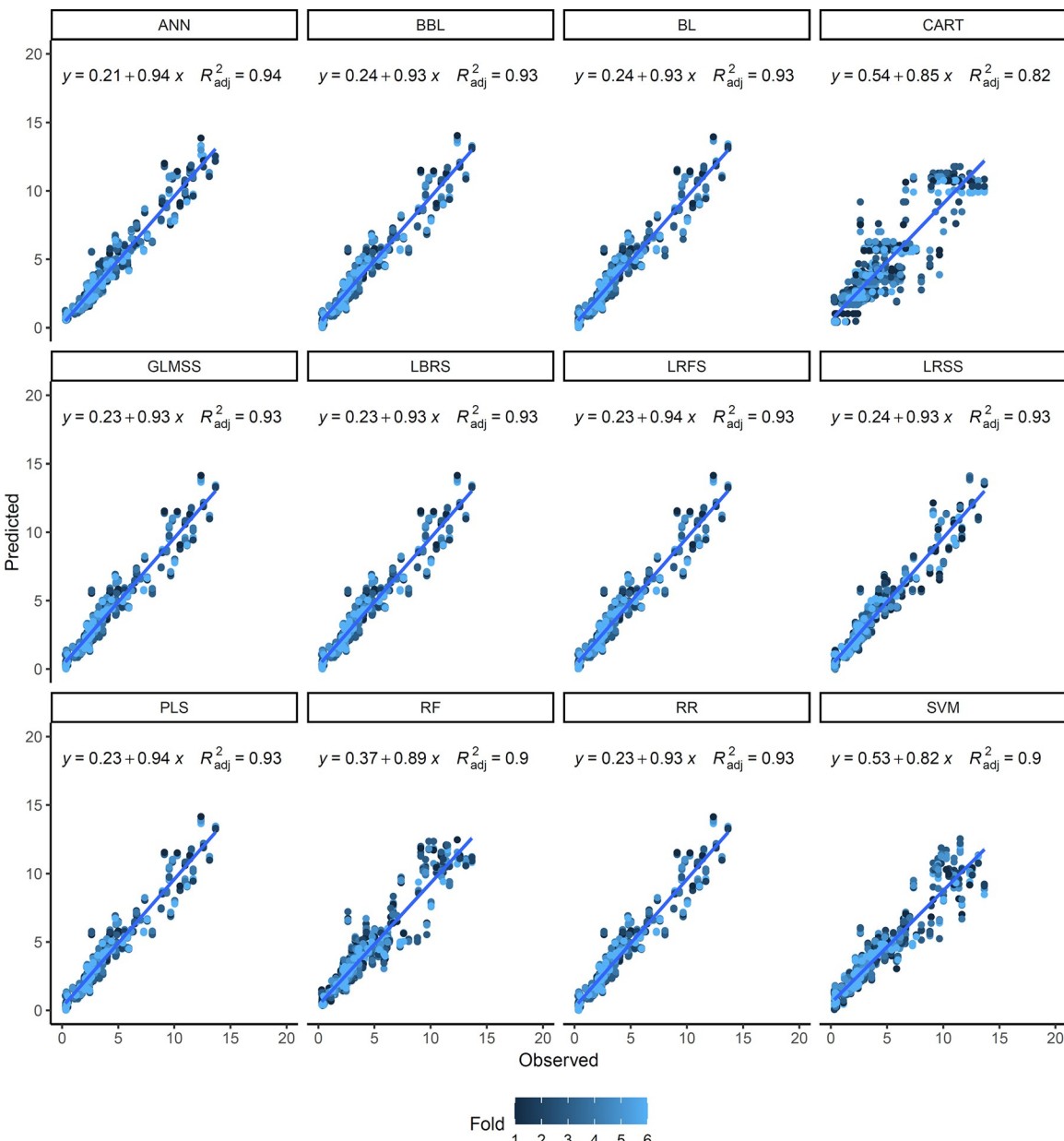

**Fig 6. Relationship between observed and predicted values for total carotenoid content of cassava roots.** The prediction was performed based on 12 different models based on random cross-validation without test set (V-Random) (80/20% in training and validation, respectively). Artificial neural network (ANN), Bayesian Blasso (BBL), Bayesian Lasso (BL), classification and regression trees (CART), generalized linear model with stepwise feature selection (GLMSS), linear regression with backward selection (LRBS), linear regression with forward selection (LRFS), linear regression with stepwise selection (LRSS), partial least squares (PLS), random forest (RF), ridge regression (RR), and support vector machine (SVM) were calculated. Each cross-validation fold is represented in different colors. The numerical data included in the graphs represent: linear equations (y) and coefficient of determination ($R^2$).

TCC varied from 1.02 to 10.40 μg g$^{-1}$ [38]. However, in breeding populations specifically aimed at increasing TCC, this variation may be even wider, such as the variation from 0.2 μg g$^{-1}$ to 25.5 μg g$^{-1}$ observed by [5].

The variation observed in the TCC constitutes a wide range in which to build predictive models with high accuracy for this trait. Therefore, this study analyzed the potential use of

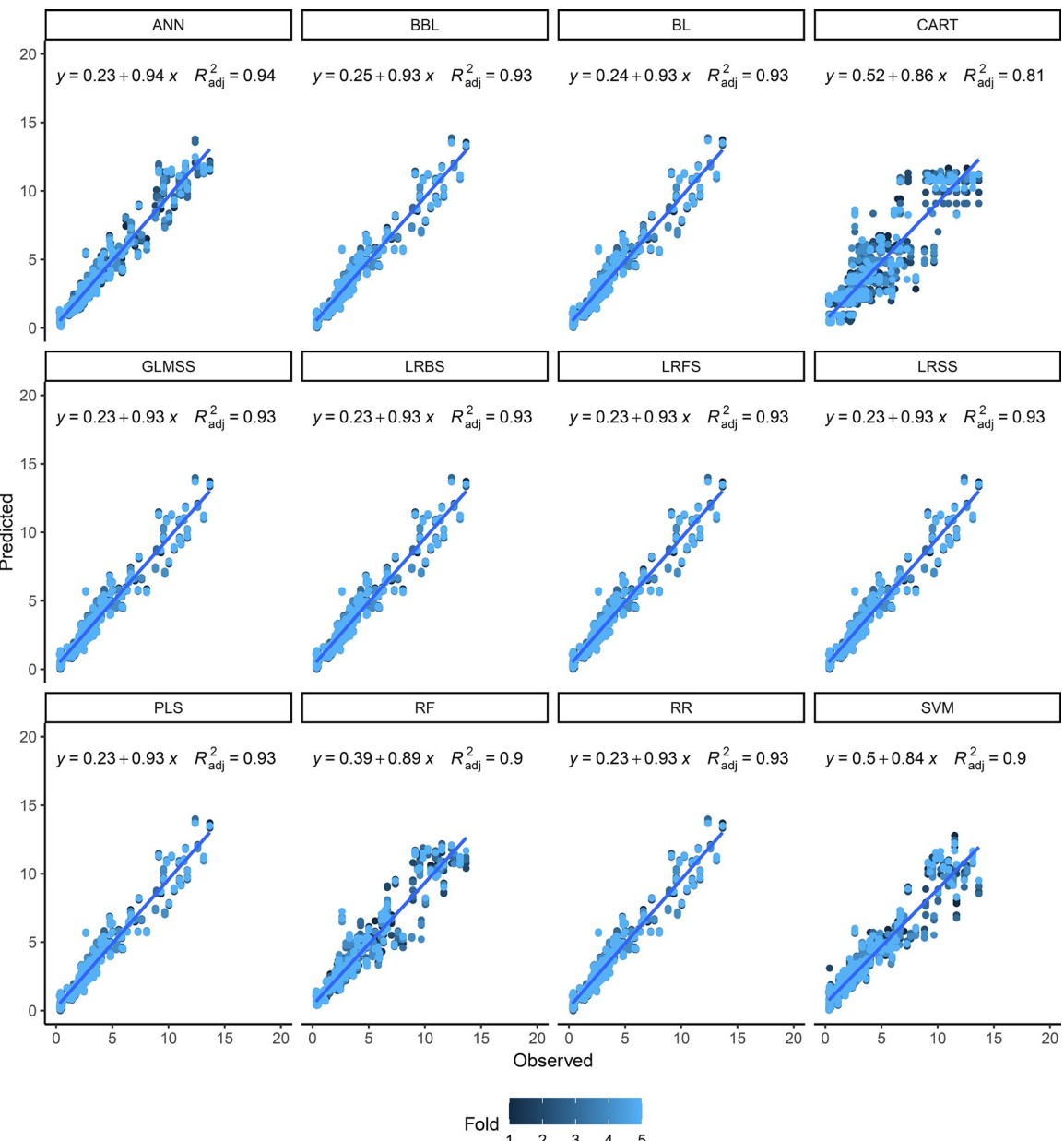

**Fig 7. Relationship between observed and predicted values for total carotenoid content of cassava roots.** The prediction was performed based on 12 different models based on random cross-validation with test set (IV-Random) (60/20/20% in training, validation, and test set respectively). Artificial neural network (ANN), Bayesian Blasso (BBL), Bayesian Lasso (BL), classification and regression trees (CART), generalized linear model with stepwise feature selection (GLMSS), linear regression with backward selection (LRBS), linear regression with forward selection (LRFS), linear regression with stepwise selection (LRSS), partial least squares (PLS), random forest (RF), ridge regression (RR), and Support vector machine (SVM) were calculated. Each cross-validation fold is represented in different colors. The numerical data included in the graphs represent: linear equations (y) and coefficient of determination ($R^2$).

rapid phenotyping based on digital images and the ability to estimate TCC in cassava roots from colorimetric data using the CIELAB color system [39, 40].

The presence of carotenoid pigments contributes to the intensity of the pulp color of cassava roots [10, 14], providing a positive correlation between these two traits [40]. Thus, first, the relationship between TCC and variables related to color was evaluated, with a strong

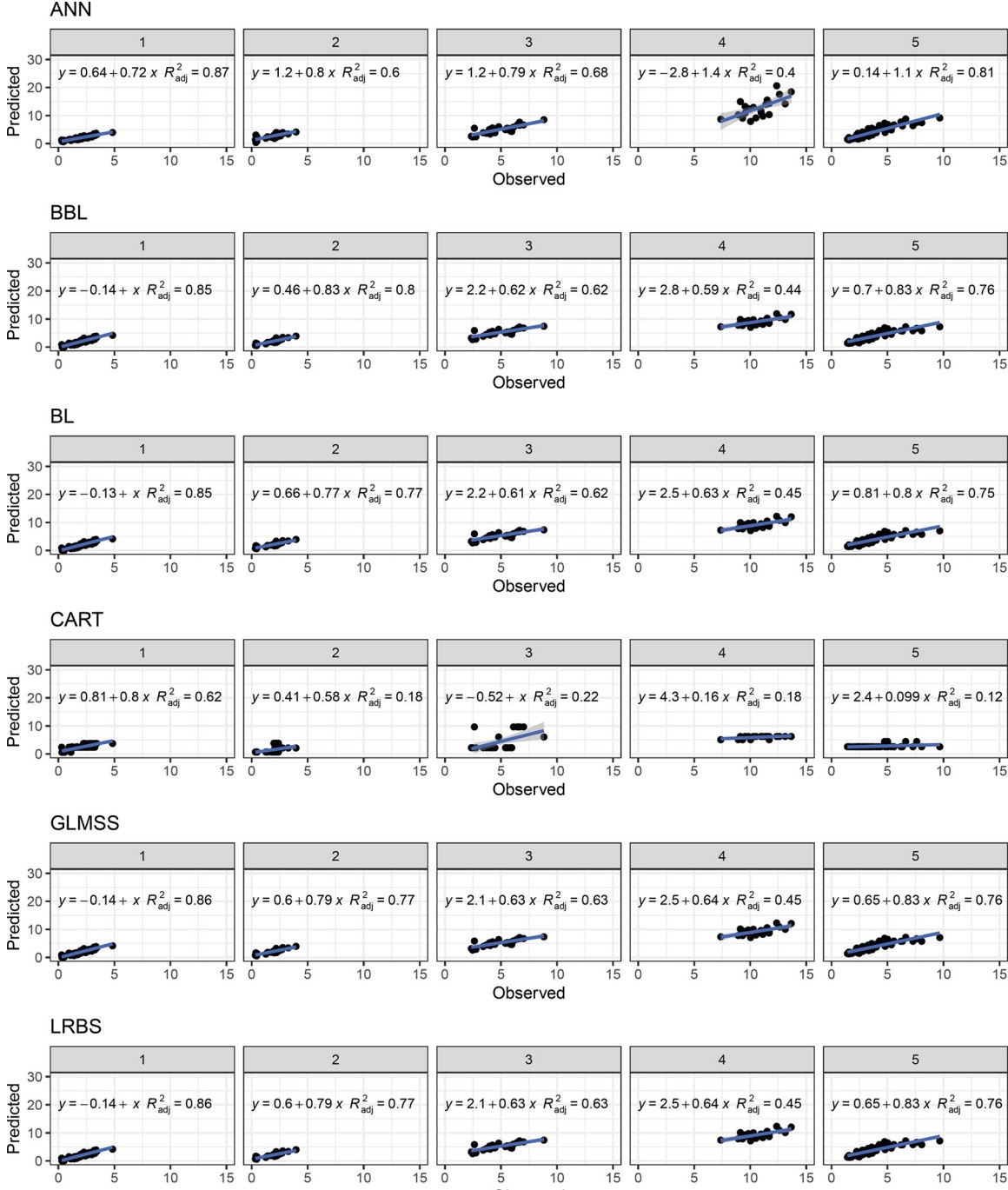

**Fig 8. Relationship between observed and predicted values for total carotenoid content of cassava roots.** The prediction was performed based on 12 different models based on PCA clustering-based with k = 5 (IV-Cluster). Artificial Neural Network (ANN), Bayesian Blasso (BBL), Bayesian Lasso (BL), Classification and Regression Trees (CART), Generalized Linear Model with Stepwise Feature Selection (GLMSS), and Linear Regression with Backward Selection (LRBS). The numerical data included in the graphs represent: linear equations (y) and coefficient of determination ($R^2$).

correlation observed between TCC and $b^*$ (r = 0.90). This strong correlation was expected as the positive values of the $b^*$ axis correspond to yellow color. This result is in agreement with studies of other species, in which it was verified that the accumulation of carotenoids indicates

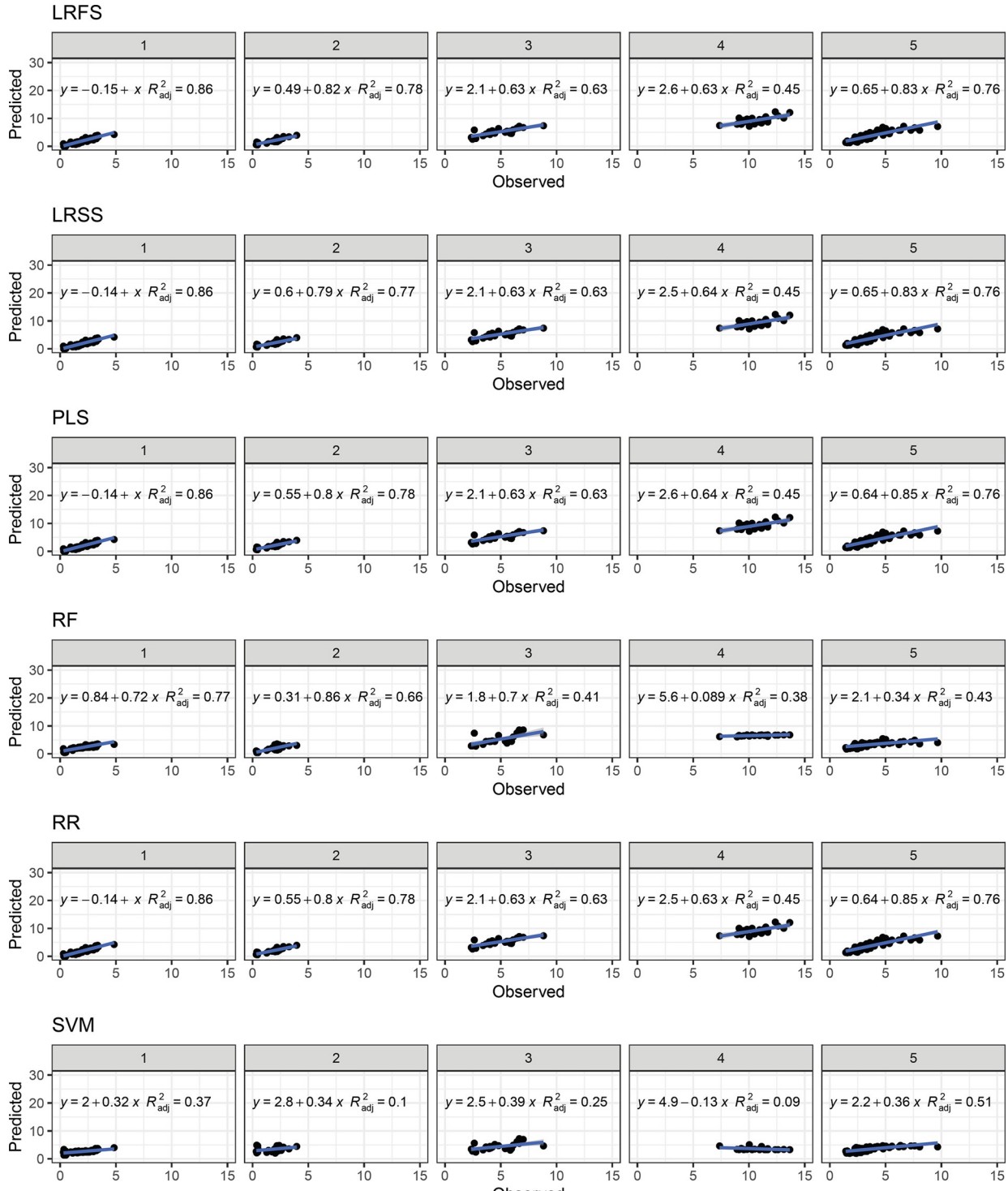

**Fig 9. Relationship between observed and predicted values for total carotenoid content of cassava roots.** The prediction was performed based on 12 different models based on PCA clustering-based with k = 5 (IV-Cluster). Linear Regression with Forward Selection (LRFS), Linear Regression with Stepwise Selection (LRSS), Partial Least Squares (PLS), Random Forest (RF), Ridge Regression (RR), and Support Vector Machine (SVM). The numerical data included in the graphs represent: linear equations (y) and coefficient of determination ($R^2$).

a tendency to increase $b^*$, as verified in several Kinnow mandarin varieties (*Citrus reticulata* Blanco) [24] and in extruded corn samples [41]. In cassava, it has also been found that higher $b^*$ values were associated with cassava samples with higher carotenoid content [40]. This behavior is due to the correlation between the $b^*$ parameter and β-carotene [42], since β-carotene is the most abundant carotenoid in cassava [10].

The same pattern of correlation was observed between TCC and chroma (R = 0.89) (Fig 1). The chroma parameter indicates the color intensity [25]; therefore, the higher the TCC, the stronger the color intensity of the cassava pulp. Similar correlations have been found between $b^*$ and chroma when analyzing the individual carotenoids content and TCC in pumpkins (*Cucurbita* spp.) [39]. In addition, these authors observed that the hue angle was negatively correlated with TCC (r = -0.83), similarly to what was observed in the present study in cassava (r = -0.59), suggesting that as the angles decrease, carotenoid concentrations increase. Furthermore, the authors of [39] identified strong correlations between colorimetric indices and carotenoid content, and stated that these correlations can be useful in indirect selection for high carotenoid content in improved pumpkin populations, being simultaneously easy to implement and low cost.

## Clustering the phenotypic diversity of total carotenoid content based on colorimetric indices

The two main components explained almost 90% of the phenotypic variance, and thus were able to demonstrate the dispersion of cassava genotypes and the formation of five groups with similar characteristics in terms of TCC and colorimetric indices. Specifically, groups 3 and 4 gathered accessions with yellow pulp and consequently higher levels of TCC, $b^*$ and chroma, since these two colorimetric indices were positively correlated with TCC.

The potential for clustering genotypes based on colorimetric indices has been recently analyzed in several species, including in cassava. The authors of [40] evaluated the clustering potential of different samples of cassava roots based on CIELAB color space indices, obtained by means of a colorimeter. The authors found that samples with higher TCC had higher $b^*$ values. They also reported clustering of genotypes via PCA based on root pulp color and TCC, using colorimetric data.

In [43], researchers used PCA and other clustering methods from the spectrophotometric data matrix applied in the UV-Vis region (400–500nm) in cassava root samples with cream, yellow and reddish pulp colors. In this strategy, the first two components represented 99.97% of the variance, clearly revealing three groups according to the carotenoid contents. These authors attribute the grouping of genotypes to the carotenoids content and the discrepancy of the reddish pulp genotype to the presence of lycopene in relevant quantities, detected by chromatographic analysis.

The efficiency of PCA in distinguishing different samples was also described in [44], which reported the morphometric and colorimetric diversity of fruits from the Balkan pepper collection (*Capsicum* spp.) and performed grouping via PCA. One of the characteristics evaluated that most contributed to the variability of peppers was the fruit color, estimated by the CIELAB parameters, obtained by the Tomato Analyzer software. The similarities between pomegranate (*Punica granatum* L.) varieties in antioxidant activity and physiochemical fruit properties have also been reported [45]. These authors verified the significant contribution to the PCA grouping of the $a^*$ index measured in the fruit juice and noted its correlation with other important attributes for the crop, such as the total phenolic content and the total anthocyanin content.

## Performance of TCC prediction models

We used different cross-validation approaches (with and without population testing and PCA clustering) to verify the predictions' accuracy. The main findings showed that it is necessary to include the large phenotype amplitude in the training and validation populations, to enable obtaining highly accurate predictions, regardless of the model used. Therefore, cluster-based and cross-validation approaches to genotypes with contrasting TCC did not fulfill this premise, in general tending to exhibit low prediction accuracies.

The various prediction models based on machine learning showed high predictive ability for TCC in cassava roots, indicated by the high $R^2$ and low RMSE values. In particular, the model based on ANN was the most accurate, providing a better fit for RMSE (0.24) and $R^2$ (0.94). Similar results were also observed in [23], which identified the high predictive ability of carotenoid content in olive oil (*Olea europaea*) using ANN ($R^2 = 0.94$) and SVM ($R^2 = 0.96$). In mandarins (*Citrus reticulata* Blanco), the PLS ($R^2 = 0.96$) and third-order linear regression ($R^2 = 0.95$) models were able to predict ripeness (citrus color index) with high accuracy [24]. Additionally, a methodology has been presented for predicting the individual carotenoid content from digital image analysis parameters of pollen samples of species from the Brassicaceae, Myrtaceae, and Fabaceae families, based on multiple linear regression, with high $R^2$ (ranging from 0.76 to 0.89) [31].

ANN models use techniques inspired by the brain and the manners in which it learns and processes information [46]. The ANN model, like the others, uses supervised learning, which aims to provide the prediction of an output variable according to known input variables [47]. Machine learning models are an efficient tool and have been widely used in different agriculture approaches to unravel, quantify, and understand data-intensive processes [47]. However, the predictive ability depends on the dataset under analysis; for this reason, it is interesting to evaluate different approaches in order to explore differences in algorithms that may benefit the predictions [48].

The search for high-throughput phenotyping methodologies has been a priority in cassava breeding programs. Therefore, several groups have been searching for alternatives to shorten selection cycles, increase the number of evaluated genotypes, and reduce costs associated with the phenotyping that are difficult to measure, such as carotenoid content. Even with the implementation of rapid phenotyping methodologies, few studies have been dedicated to the validation of models that predict traits associated with nutritional quality, such as dry matter content and carotenoids in cassava, based on image analysis. Models with good predictive accuracy can facilitate early selection of only the most interesting genotypes, allowing selection optimization.

The use of easy-to-measure predictive variables, such as colorimetric data extracted from digital images, is a key factor in accelerating the evaluation of genotypes in breeding programs aimed at screening thousands of individuals annually. This approach demonstrated that the variables $b^*$ and chroma were strongly correlated with TCC in cassava roots (Fig 1). This result makes it possible to further simplify the phenotyping protocol for TCC in cassava roots using digital images, allowing for the reduction of variables without the loss of predictive ability.

Previous authors have also used colorimetric techniques to predict TCC in cassava roots. It has been demonstrated that color measurement from colorimetric data using the CIELAB color system can be used as a fast and non-destructive method to calibrate the TCC of roots with acceptable prediction error [40]. The authors used CIELAB data and TCC data determined by spectrophotometry as input to various machine learning models and found $R^2$ ~ 0.60. Similarly, the authors of [43] used the colorimetric technique associated with UV-Vis/ HPLC to predict TCC in cassava roots, finding $R^2 > 0.40$ and RMSE < 9.99 for the PLS, SVM,

and Elastic Net models. Therefore, the approaches used in this study allowed the obtention of greater predictive power and smaller error compared to similar reports in the literature. A possible explanation for this is the fact that colorimeters were used in [40] and [43] to obtain the CIELAB colorimetric indices. It is speculated that digital images allow for greater precision in color analysis, as they capture all the pixels of root images, while colorimeters measure only a specific portion of the sample.

The software used in the present study for analyzing RGB images, Tomato Analyzer, has also showed high potential for quantifying characteristics that are difficult to measure by conventional phenotyping methods, such as fruit wall thickness and pericarp thickness in pepper fruits (*Capsicum* spp.) evaluated in germplasm collected in different regions of the Balkans [44]. The Tomato Analyzer allows the estimation of a large number of characteristics associated with the shape and colors derived from digitized images of fruit sections [49] or even roots, as was the case in the current study.

In general, the development of predictive models based on different approaches has shown its great utility and reliability in estimating TCC in cassava. One of the first studies to predict the carotenoids content in cassava was developed based on NIRS data, resulting in predictive abilities of 0.92 and 0.93 for TCC and β-carotene, respectively, using PLS regression [5]. In another study, also using NIRS data, the practical use of carotenoid content predictions for selection in species breeding programs was demonstrated for the first time [15]. Based on the use of LOCAL regression, these authors reported $R^2 = 0.74$ for predictions of the β-carotene content. More recently, NIRS data has been combined with different predictive models [16, 17], such as the modified partial least squares and RF approaches. These last authors reported a high predictive ability for β-carotene content ($R^2 = 0.99$).

In the present study, phenotyping via digital images also demonstrated its high predictive ability ($R^2 = 0.94$ and RMSE = 0.24) for TCC prediction based on the ANN model. Therefore, this offers a new approach for measuring this trait in cassava, with much lower costs in the acquisition and maintenance of equipment. Already, the implementation of a computer vision system based on RGB images to detect colorimetric characteristics in tomatoes, using low-cost equipment, has showed excellent classification capacity [50]. An application has also been developed [27] to quickly determine the lycopene content in tomato fruits and classify the different stages of maturation based on color, based on $L^*$, $a^*$, and $b^*$ parameters, obtained from RGB images. Therefore, the use of digital image analysis techniques, based on free software and low-cost cameras, enables the construction of accurate and reliable predictive models for the quantification of TCC in cassava. These aspects optimize the analysis time and human and financial resources required without sacrificing the reliability of the results.

## Future perspectives

The potential for implementing rapid phenotyping based on digital images for TCC prediction was demonstrated in this study. In addition, image analysis allows the analysis and incorporation of other characteristics of agronomic importance in cassava cultivation, such as the evaluation of canopy and root yield-related traits, assessment of the linkage between root architecture and micronutrient (zinc and calcium) concentration, and estimation of the growth and nutritional performance of cassava under poor irrigation and potassium fertigation [51–53]. Therefore, these variables can be used in the composition of multi-trait selection indexes for further optimization of cassava breeding programs. Furthermore, the joint use of this approach with genomic selection tools can accelerate the development of cassava varieties with higher TCC.

This study completed the first step in the construction of a digital image database to predict TCC in cassava. The next steps will involve the inclusion of root samples with higher TCC to aid the reduction of predictive error, particularly in samples exceeding 5.0 μg g$^{-1}$. In addition, samples of roots grown in different cultivation environments will be evaluated to include the effect of genotype × environment interaction in the predictive models so that they can achieve more widespread use in different breeding programs. Such improvements are predominantly focused on increasing the predictive ability of TCC and the practical implementation of high-throughput phenotyping for screening hundreds of samples daily. The expectation is that image-based phenotyping methodologies will improve the quality and accuracy of data collected in various phenotypic evaluations, thus contributing to an increase in the selection differential and in the heritability coefficient, with a direct effect on genetic gain [20].

## Conclusion

The use of colorimetric data from the CIELAB space, obtained from the analysis of RGB digital images, is an economical, fast, and efficient alternative that has shown an excellent predictive ability for the TCC in cassava roots. The $b^*$ and chroma indices of the CIE color model are strongly correlated with TCC and can be used to accurately assess this characteristic in cassava. These indices were also the variables that most influenced the accuracy of predictive models, and that most contributed to clustering genotypes with higher TCC levels based on PCA.

The ANN model showed the greatest predictive ability of all the algorithms for TCC in cassava roots, even producing accuracy and prediction errors comparable to more sophisticated approaches such as the NIRS. Thus, this first proof of concept demonstrated the high potential of using digital images for the practical implementation of high-throughput phenotyping in cassava.

## Supporting information

**S1 Table. List of cassava genotypes and the total carotenoid content and colorimetric indices.**
(CSV)

**S2 Table. Observed and predicted values for total carotenoid content of cassava roots.**
based on Linear Regression with Forward Selection (LRFS). Linear Regression with Backwards Selection (LRBS). Ridge Regression (RR). Linear Regression with Stepwise Selection (LRSS). Generalized Linear Model with Stepwise Feature Selection (GLMSS). Random Forest (RF). Partial Least Squares (PLS). the Bayesian Lasso (BL). the Bayesian Blasso (BBL). Support vector machine (SVM). and Classification. regression trees (CART) and Artificial Neural Network (ANN). The cross-validation scheme was performed with 5 repetitions of 6-fold analysis.
(CSV)

**S1 Fig. Boxplot of the performance of the random cross-validation without test set (V-Random) to predict the total carotenoid content in cassava roots using colorimetric indices obtained from digital images considering the complete (all variables) and reduced models (only variables with more than 50% relative importance).** Artificial Neural Network (ANN), Bayesian Blasso (BBL), Bayesian Lasso (BL), Classification and Regression Trees (CART), Generalized Linear Model with Stepwise Feature Selection (GLMSS), Linear Regression with Backward Selection (LRBS), Linear Regression with Forward Selection (LRFS), Linear Regression with Stepwise Selection (LRSS), Partial Least Squares (PLS), Random Forest (RF), Ridge Regression (RR), and Support Vector Machine (SVM).
(TIFF)

**S2 Fig. Performance of PCA clustering-based cross-validation scheme (IV-Cluster) to predict the total carotenoid content in cassava roots using colorimetric indices obtained from digital images considering the complete (all variables) and reduced model (only variables with more than 50% relative importance).** Artificial Neural Network (ANN), Bayesian Blasso (BBL), Bayesian Lasso (BL), Classification and Regression Trees (CART), Generalized Linear Model with Stepwise Feature Selection (GLMSS), Linear Regression with Backward Selection (LRBS), Linear Regression with Forward Selection (LRFS), Linear Regression with Stepwise Selection (LRSS), Partial Least Squares (PLS), Random Forest (RF), Ridge Regression (RR), and Support Vector Machine (SVM).
(TIFF)

**S3 Fig. Boxplot of the performance of the random cross-validation with test set (IV-Random) to predict the total carotenoid content in cassava roots using colorimetric indices obtained from digital images considering the complete (all variables) and reduced model (only variables with more than 50% relative importance).** Artificial Neural Network (ANN), Bayesian Blasso (BBL), Bayesian Lasso (BL), Classification and Regression Trees (CART), Generalized Linear Model with Stepwise Feature Selection (GLMSS), Linear Regression with Backward Selection (LRBS), Linear Regression with Forward Selection (LRFS), Linear Regression with Stepwise Selection (LRSS), Partial Least Squares (PLS), Random Forest (RF), Ridge Regression (RR), and Support Vector Machine (SVM).
(TIFF)

## Author Contributions

**Conceptualization:** Ravena Rocha Bessa de Carvalho, Massaine Bandeira e Sousa, Eder Jorge de Oliveira.

**Data curation:** Ravena Rocha Bessa de Carvalho, Massaine Bandeira e Sousa.

**Formal analysis:** Diego Fernando Marmolejo Cortes, Massaine Bandeira e Sousa.

**Funding acquisition:** Eder Jorge de Oliveira.

**Investigation:** Ravena Rocha Bessa de Carvalho, Diego Fernando Marmolejo Cortes, Massaine Bandeira e Sousa.

**Methodology:** Diego Fernando Marmolejo Cortes, Massaine Bandeira e Sousa, Eder Jorge de Oliveira.

**Project administration:** Eder Jorge de Oliveira.

**Resources:** Eder Jorge de Oliveira.

**Supervision:** Luciana Alves de Oliveira, Eder Jorge de Oliveira.

**Writing – original draft:** Ravena Rocha Bessa de Carvalho, Diego Fernando Marmolejo Cortes, Massaine Bandeira e Sousa.

**Writing – review & editing:** Luciana Alves de Oliveira, Eder Jorge de Oliveira.

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
