## [Decision Letter · Decision Letter 0]

2 Nov 2021

PONE-D-21-31303Image-based phenotyping of cassava roots for diversity studies and carotenoids predictionPLOS ONE

Dear Dr. de Oliveira,

Thank you for submitting your manuscript to PLOS ONE. After careful consideration, we feel that it has merit but does not fully meet PLOS ONE’s publication criteria as it currently stands. Therefore, we invite you to submit a revised version of the manuscript that addresses the points raised during the review process.

Please address the concerns of both reviewers, especially the model validation with independent dataset.==============================

We look forward to receiving your revised manuscript.

Kind regards,

Peng Zhang, Ph.D.

Academic Editor

PLOS ONE

Journal Requirements:

When submitting your revision, we need you to address these additional requirements. 1. Please ensure that your manuscript meets PLOS ONE's style requirements, including those for file naming. The PLOS ONE style templates can be found at https://journals.plos.org/plosone/s/file?id=wjVg/PLOSOne_formatting_sample_main_body.pdf and https://journals.plos.org/plosone/s/file?id=ba62/PLOSOne_formatting_sample_title_authors_affiliations.pdf. 2. Thank you for stating the following in the Acknowledgments Section of your manuscript: [The authors thank Conselho Nacional de Desenvolvimento Científico e Tecnológico, Fundação de Amparo à Pesquisa do Estado da Bahia, and Coordenação de Aperfeiçoamento de Pessoal de Nível Superior for financial support. This work was also supported by the NEXTGEN Cassava project, through a grant to Cornell University by the UK’s Foreign, Commonwealth & Development Office (FCDO) and the Bill & Melinda Gates Foundation (Grant INV-007637 http://www.gatesfoundation.org).] We note that you have provided funding information that is not currently declared in your Funding Statement. However, funding information should not appear in the Acknowledgments section or other areas of your manuscript. We will only publish funding information present in the Funding Statement section of the online submission form. Please remove any funding-related text from the manuscript and let us know how you would like to update your Funding Statement. Currently, your Funding Statement reads as follows:  [●
Ravena Rocha Bessa de Carvalho: CAPES (Coordenação de Aperfeiçoamento de Pessoal de Nível Superior). Grant number:88882.424436/2019-01●
Eder Jorge de Oliveira: CNPq (Conselho Nacional de Desenvolvimento Científico e Tecnológico). Grant number:409229/2018-0, 442050/2019-4 and 303912/2018-9●
Eder Jorge de Oliveira: FAPESB (Fundação de Amparo à Pesquisa do Estado da Bahia). Grant number: Pronem 15/2014●
Eder Jorge de Oliveira and Massaine Bandeira e Sousa: UK’s Foreign, Commonwealth & Development Office (FCDO) and the Bill & Melinda Gates Foundation. Grant number: INV-007637●
The funder provided support in the form of fellowship and funds for the research, but did not have any additional role in the study design, data collection and analysis, decision to publish, or preparation of the manuscript.] Please include your amended statements within your cover letter; we will change the online submission form on your behalf.

Reviewers' comments:

Reviewer's Responses to Questions

**Comments to the Author**

1. Is the manuscript technically sound, and do the data support the conclusions?

Reviewer #1: Yes

Reviewer #2: Yes

2. Has the statistical analysis been performed appropriately and rigorously? 

Reviewer #1: Yes

Reviewer #2: Yes

3. Have the authors made all data underlying the findings in their manuscript fully available?

Reviewer #1: Yes

Reviewer #2: Yes

4. Is the manuscript presented in an intelligible fashion and written in standard English?

Reviewer #1: Yes

Reviewer #2: Yes

5. Review Comments to the Author

Reviewer #1: This study aimed to use digital images to extract information on the pulp color of cassava roots and estimate correlations with TCC, and select predictive models for TCC using colorimetric indices. The results showed good correlation between TCC and b and chroma, and the predictive ability of ANN could achieve R2 with 0.94. In general, this paper is well written and organized. Before accept, several minor issues should be addressed.

1. From line 31 to line 34, using two sentences to describe the methods of data analysis and prediction models is cumbersome, and you can combine these two sentences.

2. In line 121, select the standard root for image collection. What is the standard root defined?

3. In Fig 1., how to understand the shaded area next to the blue line?

4. In Fig 5., ordinate of each figure use “L., a., b.” which is not accurate, it should be in line with “L*, a*, b*”. What’s more, abscissa of each figure should be added with units (%), or it should be explained in the notes of the figure.

5. In Table 1., the column of “Modelo” might be “Model”.

6. In line 265, “performed similarly (RMSE = 0.23, R2 = 0.93)” should be “RMSE = 0.26”.

7. From line 265 to 267, “…had high predictive abilities (R2 = 0.91 and 0.90, respectively)”, the values of R2 come from different model (Complete model and Reduced model, respectively). But “they also had higher RMSE values (0.32 and 0.33, respectively)”, the values of RMSE come from the same model (Reduced model). How should we understand it here?

8. In Fig 6. and its notes, there is no description of total carotenoid content units.

9. What is the significance of the simplified model? Compared with the complete model, the prediction effect of simplified model is not significantly improved, and the prediction ability of some models is even decreased.

10. Is the root that was ground into powder the same sample as the root that collected image information? If it is, during the process from root to powder (including the image acquisition process), the root is exposed to air, is the TCC content stable?

11. The MAPE of each model is suggested to add in the main content.

Reviewer #2: This study used RGB images of cassava roots to predict total carotenoids contents with a 228 genotypes population from cassava germplasm. Chromatic components L*, a* and b* as well as hue and lightness were calculated from the RGB values with the CIELAB system. As the correlation between each of these parameters, eg. L*, a*, b*, hue and lightness with total carotenoids contents is not very high, so the author tried to develop models based on these parameters (eg. L*, a*, b*, hue and lightness) to predict the total carotenoids contents and got very high correlation coefficients. Especially, this study used 12 different models.

However, I have two major concerns.

First, I did not find the model validation with independent dataset. I suggest the author used part of the data to build model and use the other part of the data to validate the models.

Second, how the PCA analysis and k-means clustering necessary or useful in logic to developing the prediction models? I see that the 228 genotypes can be perfectly clustered into 5 clusters by the phenotypes, so I suggest to use different clusters to develop models and use the left clusters to validate. I assume that the models developed based on different clusters should have different predictivity.

The methods and equations for calculating L*, a* and b* were not presented in the Methods.

L137-143, the equations should be presented in separate rows, not mixed in text. This should be applied through the manuscript.

Fig1 legend, the blue line is not for 1:1.

Fig6 legend, the blue line is not for 1:1 isoline from the figure.

6. PLOS authors have the option to publish the peer review history of their article (what does this mean?). If published, this will include your full peer review and any attached files.

Reviewer #1: No

Reviewer #2: No

---

## [Author Response · Author response to Decision Letter 0]

15 Dec 2021

Reviewer #1: 

1. From line 31 to line 34, using two sentences to describe the methods of data analysis and prediction models is cumbersome, and you can combine these two sentences.

Response: There was an error here. The two sentences were duplicated. Thank you. 

2. In line 121, select the standard root for image collection. What is the standard root defined?

Response: We clarified this sentence: 

“During harvest, 4–6 commercial roots (>5 cm in diameter and around 12–20 cm in length) of all 228 genotypes were selected.”

3. In Fig 1., how to understand the shaded area next to the blue line?

Response: It is the 95% confidence region. We changed the caption of Figure 1. 

4. In Fig 5., ordinate of each figure use “L., a., b.” which is not accurate, it should be in line with “L*, a*, b*”. What’s more, abscissa of each figure should be added with units (%), or it should be explained in the notes of the figure.

Response: We changed the figures according to reviewer’s suggestion.

5. In Table 1., the column of “Modelo” might be “Model”.

Response: Ok, thanks.

6. In line 265, “performed similarly (RMSE = 0.23, R2 = 0.93)” should be “RMSE = 0.26”.

Response: Ok, thanks.

7. From line 265 to 267, “…had high predictive abilities (R2 = 0.91 and 0.90, respectively)”, the values of R2 come from different model (Complete model and Reduced model, respectively). But “they also had higher RMSE values (0.32 and 0.33, respectively)”, the values of RMSE come from the same model (Reduced model). How should we understand it here?

Response: We tried to clarify the sentence as:

“Although the RF and SVM models had high predictive abilities (R2 = 0.90 for both complete and reduced models), they also had higher RMSE values (0.31 and 0.32 for complete and reduced model, respectively (RF) and 0.33 for both complete and reduced (SVM).”

8. In Fig 6. and its notes, there is no description of total carotenoid content units.

Response: We added the unit (ug.g-1). Thanks. 

9. What is the significance of the simplified model? Compared with the complete model, the prediction effect of simplified model is not significantly improved, and the prediction ability of some models is even decreased.

Response: The paired t-test was performed based on repeated k-folds cross-validation to compare the prediction algorithms of the complete and reduced models.

10. Is the root that was ground into powder the same sample as the root that collected image information? If it is, during the process from root to powder (including the image acquisition process), the root is exposed to air, is the TCC content stable?

Response: The roots used for capturing imagens were the same grounded for the TCC analysis. Although the carotenoids are sensitive to ultraviolet light and high temperature, we adopted different strategies to minimize any adverse changes in TCC due to such effects protecting them from UV light (by covering the samples with aluminum foil to avoid contact with light, whenever possible) and avoiding excessively high temperatures using environment with controlled temperature. Please see the changes in Material and methods. 

11. The MAPE of each model is suggested to add in the main content.

Response: We added the mean absolute percentage error 

Reviewer #2:

This study used RGB images of cassava roots to predict total carotenoids contents with a 228 genotypes population from cassava germplasm. Chromatic components L*, a* and b* as well as hue and lightness were calculated from the RGB values with the CIELAB system. As the correlation between each of these parameters, eg. L*, a*, b*, hue and lightness with total carotenoids contents is not very high, so the author tried to develop models based on these parameters (eg. L*, a*, b*, hue and lightness) to predict the total carotenoids contents and got very high correlation coefficients. Especially, this study used 12 different models.

However, I have two major concerns.

First, I did not find the model validation with independent dataset. I suggest the author used part of the data to build model and use the other part of the data to validate the models.

Second, how the PCA analysis and k-means clustering necessary or useful in logic to developing the prediction models? I see that the 228 genotypes can be perfectly clustered into 5 clusters by the phenotypes, so I suggest to use different clusters to develop models and use the left clusters to validate. I assume that the models developed based on different clusters should have different predictivity.

Response: We added three cross-validation approaches to get the accuracy, MAPE and RMSE parameters. Please see in “Material and methods” and “Results” sections. 

The methods and equations for calculating L*, a* and b* were not presented in the Methods.

Response: The equations for converting RGB values into CieLab values were included in the manuscript

L137-143, the equations should be presented in separate rows, not mixed in text. This should be applied through the manuscript.

Response: Ok!

Fig1 legend, the blue line is not for 1:1.

Response: There was an error here. The blue line represents the regression line.

Fig6 legend, the blue line is not for 1:1 isoline from the figure.

Response: The same error of Figure 1. We have changed. Thanks.

---

## [Decision Letter · Decision Letter 1]

17 Jan 2022

Image-based phenotyping of cassava roots for diversity studies and carotenoids prediction

PONE-D-21-31303R1

Dear Dr. de Oliveira,

We’re pleased to inform you that your manuscript has been judged scientifically suitable for publication and will be formally accepted for publication once it meets all outstanding technical requirements.

Kind regards,

Peng Zhang, Ph.D.

Academic Editor

PLOS ONE

Additional Editor Comments (optional):

Reviewers' comments:

Reviewer's Responses to Questions

**Comments to the Author**

1. If the authors have adequately addressed your comments raised in a previous round of review and you feel that this manuscript is now acceptable for publication, you may indicate that here to bypass the “Comments to the Author” section, enter your conflict of interest statement in the “Confidential to Editor” section, and submit your "Accept" recommendation.

Reviewer #1: All comments have been addressed

Reviewer #2: All comments have been addressed

2. Is the manuscript technically sound, and do the data support the conclusions?

Reviewer #1: Yes

Reviewer #2: Yes

3. Has the statistical analysis been performed appropriately and rigorously? 

Reviewer #1: Yes

Reviewer #2: Yes

4. Have the authors made all data underlying the findings in their manuscript fully available?

Reviewer #1: Yes

Reviewer #2: Yes

5. Is the manuscript presented in an intelligible fashion and written in standard English?

Reviewer #1: Yes

Reviewer #2: Yes

6. Review Comments to the Author

Reviewer #1: (No Response)

Reviewer #2: (No Response)

7. PLOS authors have the option to publish the peer review history of their article (what does this mean?). If published, this will include your full peer review and any attached files.

Reviewer #1: No

Reviewer #2: **Yes: **Qingfeng Song

---

## [Editor Report · Acceptance letter]

21 Jan 2022

PONE-D-21-31303R1 

Image-based phenotyping of cassava roots for diversity studies and carotenoids prediction 

Dear Dr. de Oliveira:

I'm pleased to inform you that your manuscript has been deemed suitable for publication in PLOS ONE. Congratulations! Your manuscript is now with our production department. 

Kind regards, 

on behalf of

Prof. Dr. Peng Zhang 

Academic Editor

PLOS ONE